# Large-Scale, High-Throughput Phenotyping of the Postharvest Storage Performance of ‘Rustenburg’ Navel Oranges and the Development of Shelf-Life Prediction Models

**DOI:** 10.3390/foods11131840

**Published:** 2022-06-22

**Authors:** Abiola Owoyemi, Ron Porat, Amnon Lichter, Adi Doron-Faigenboim, Omri Jovani, Noam Koenigstein, Yael Salzer

**Affiliations:** 1Department of Postharvest Science of Fresh Produce, ARO, The Volcani Institute, Rishon LeZion 7528809, Israel; abiola.owoyemi@mail.huji.ac.il (A.O.); vtlicht@volcani.agri.gov.il (A.L.); 2Robert H. Smith Faculty of Agricultural, Food and Environmental Sciences, Hebrew University of Jerusalem, Rehovot 76100, Israel; 3Genomics and Bioinformatics Unit, ARO, The Volcani Institute, Rishon LeZion 7528809, Israel; adif@volcani.agri.gov.il; 4Department of Industrial Engineering, Tel Aviv University, P.O. Box 39040, Tel Aviv 6997801, Israel; omrijovani@gmail.com (O.J.); noamk@tauex.tau.ac.il (N.K.); 5Department of Growing, Production and Environmental Engineering, ARO, The Volcani Institute, Rishon LeZion 7528809, Israel; salzer@volcani.agri.gov.il

**Keywords:** citrus, intelligent logistics, modeling, orange, postharvest

## Abstract

We conducted a large-scale, high-throughput phenotyping analysis of the effects of various pre-harvest and postharvest features on the quality of ‘Rustenburg’ navel oranges, in order to develop shelf-life prediction models to enable the use of the First Expired, First Out logistics strategy. The examined pre-harvest features included harvest time and yield, and the examined postharvest features included storage temperature, relative humidity during storage and duration of storage. All together, we evaluated 12,000 oranges (~4 tons) from six different orchards and conducted 170,576 measurements of 14 quality parameters. Storage time was found to be the most important feature affecting fruit quality, followed by storage temperature, harvest time, yield and humidity. The examined features significantly affected (*p* < 0.001) fruit weight loss, firmness, decay, color, peel damage, chilling injury, internal dryness, acidity, vitamin C and ethanol levels, and flavor and acceptance scores. Four regression models were evaluated for their ability to predict fruit quality based on pre-harvest and postharvest features. Extreme gradient boosting (XGBoost) combined with a duplication approach was found to be the most effective approach. It allowed for the prediction of fruit-acceptance scores among the full data set, with a root mean square error (RMSE) of 0.217 and an R^2^ of 0.891.

## 1. Introduction

Orange (*Citrus sinensis*) is the world’s fifth largest fruit crop (after banana, watermelons, apples and grapes), with an annual global production of 75.4 million tons [1]. Oranges, similar to other citrus fruits, are very popular and consumers appreciate their delicate, fruity and refreshing flavor, as well as their high nutritional value [2].

‘Rustenburg’ navel orange, which originated in South Africa, is a high-quality late-season orange that is seedless, relatively easy to peel, has a rich fruity flavor and is excellent for fresh consumption [3]. For these reasons, it is a widely cultivated late-season cultivar in several citrus-growing countries, including Israel, and ‘Rustenburg’ fruits are commercially stored for relatively long periods of up to 3–4 months, in order to extend the marketing season [4].

The postharvest storage performance of oranges may be affected by various pre-harvest and postharvest features [5]. Pre-harvest features such as climate conditions, cultivation practices, harvest time, choice of rootstock, tree age and yield may affect postharvest quality [6,7]. For example, during ripening, the fruit undergoes continuous metabolic changes, such as the accumulation of sugars and a decrease in acidity levels, which cause changes in fruit flavor and nutritional quality over the course of the ripening period [8]. Furthermore, early-harvested oranges are relatively sensitive to postharvest chilling injury and more tolerant to postharvest decay, whereas late-harvested fruits are more tolerant to chilling, but more sensitive to microbial spoilage [9]. It has also been reported that the maturity stage at harvest may affect peel pitting during storage [10]. Other pre-harvest features, such as tree age and yield, may affect the sugar and acidity levels, firmness and physiological behaviors, manifested as respiration and the rate of ethylene production [11].

The postharvest storage performance of oranges is greatly influenced by the environmental conditions under which the fruits are stored, especially temperature and relative humidity (RH) [12]. Temperature is the most important environmental factor affecting fruit quality after harvest, as low storage temperatures reduce respiration, water loss and pathogen growth, but can also cause chilling injury (CI) [12]. The optimal RH for postharvest storage of most fruit species is between 90–95%. Lower RH levels lead to greater water loss and RH levels that are too high may lead to condensation and the enhanced growth of pathogens [13]. Overall, it is recommended that oranges be stored at 3–8 °C and 90–95% RH [12].

Currently, the postharvest storage management of fruits and vegetables is principally governed by the First In, First Out (FIFO) logistics strategy, meaning that marketing decisions are based solely on storage time irrespective of the initial quality of the produce and its remaining potential shelf life [14,15]. Although the FIFO approach is straightforward and easy to implement, it is based on the assumption that all products arriving at the cold-storage facility on a particular date have the same shelf-life potential, which all too often is not the actual case [14]. In contrast, the adoption of the more advanced First Expired, First Out (FEFO) logistics strategy would enable more efficient inventory management based on shelf-life predictions for each particular batch of produce, to ensure that only high-quality produce will reach the distinct marketing destinations [15]. However, the adoption of an intelligent FEFO logistics-management system requires the development of accurate shelf-life prediction models that will provide reliable information regarding the remaining shelf-life of each batch of produce.

In recent years, various models have been developed to predict numerous postharvest storage traits, including respiration rate, microbial growth, physical, chemical, and sensorial characteristics, and storage life [16,17,18,19]. Advanced machine learning and artificial intelligence technologies now allow the development of even more accurate forecasting and prediction models for important agriculture outputs [20,21]. Nonetheless, the development of accurate and advanced shelf-life prediction models requires the acquisition of large amounts of postharvest storage data and the consideration of all of the factors that may affect produce quality, including various pre-harvest and postharvest features [22,23].

Overall, the main objective of the current research was to conduct a large-scale, high-throughput phenotyping analysis of the postharvest storage performance of ‘Rustenburg’ navel oranges, in order to develop shelf-life prediction models to enable the implementation of the FEFO method. To that end, we evaluated 12,000 oranges (~4 tons) harvested from six different orchards and conducted 170,576 measurements of 14 quality parameters.

## 2. Materials and Methods

### 2.1. Plant Material

‘Rustenburg’ navel oranges were harvested from six commercial orchards on different dates between 21 February 2021 and 6 April 2022, as described in Table 1. The rational for choosing different harvest times was to examine the postharvest performances of fruit with different maturity indices, i.e., different total soluble solids (TSS), acidity, peel color, etc. The day after harvest, the fruits were treated in a commercial citrus packinghouse. This treatment included washing, waxing, the application of fungicides, sorting and packaging according to common commercial practices. Then, the fruits were transferred to the Department of Postharvest Science, ARO, The Volcani Institute, where they were placed in cold storage rooms as described below.

### 2.2. Postharvest Storage Conditions

Fruits from each orchard were stored at 90% RH and at 3 different storage temperatures: 2, 5 and 8 °C. Fruits from Harvest 5 (Table 1) were also stored at 5 °C under high humidity (RH = 95%) and low humidity conditions (RH = 70%). The high RH was achieved by using a PARKOO PH06LB ultrasonic humidifier (ANIA, Kfar Aviv, Israel) and the low RH was achieved by using an IVLTD08 dehumidifier (Vogel Refrigeration Services, Ltd., Rishon LeZion, Israel). The normal RH at 5 °C without modifications was ~90%. Each harvest included 60 10 kg cartons of oranges (20 cartons were stored at each temperature), while the Harvest 5 included 100 10 kg cartons of oranges (20 cartons stored at each temperature and/or RH condition). Overall, the experiment included 400 10 kg cartons of oranges (i.e., a total of 4 tons of oranges).

### 2.3. Evaluations of Fruit Quality

Evaluations of fruit quality were conducted at Time 0 and at weekly intervals over a period of 20 weeks (~4.5 months). The quality evaluations were conducted after one additional week of storage under shelf-life conditions at 20 °C. The different quality evaluations are described below.

#### 2.3.1. Firmness

Firmness was tested with a texture analyzer (TA-XT plus, Stable Micro Systems, Surrey, UK) with a 50-kg load cell, using a 75 mm (diam.) cylindrical probe. The machine compressed the samples (15 replicates for each treatment and storage time) in the equatorial zone until 5% deformation at a speed of 1 mm·s^−1^. Results were expressed as the force required to induce that level of deformation.

#### 2.3.2. Weight Loss

Weight loss was evaluated by weighing the produce at Time 0 and then again after the different storage periods. Weight-loss data are expressed as percentages of weight lost relative to the initial weight.

#### 2.3.3. Peel Color

Peel color was determined by measuring lightness (L*), chroma (C*) and hue angle (H°) values, using a Minolta Chromo Meter, Model CR-400 (Minolta, Tokyo, Japan). The presented data are means of 15 measurements.

#### 2.3.4. Peel Damage, Decay and Internal Dryness

Peel damage, the incidence of decay and internal dryness were evaluated for the different storage periods following the manual sorting of the produce. Results are expressed as the percentage of infected fruits among the total amount of produce.

#### 2.3.5. Total Soluble Solids (TSS) and Titratable Acidity (TA)

The total soluble solids (TSS) contents of the extracted juice were determined with a Model PAL-1 digital refractometer (Atago, Tokyo, Japan), and acidity levels (%) were measured by titration to pH 8.3 with 0.1 M NaOH using a Model CH-9101 automatic titrator (Metrohm, Herisau, Switzerland). Each measurement was replicated three times, with juice collected from three fruits used each time.

#### 2.3.6. Vitamin C

The vitamin C (ascorbic acid) content of the orange juice was determined by titration with 2,6-dichlorophenolindophenol, as described previously [24]. Levels of ascorbic acid were determined by comparing the titration volumes of the fruit juices with those of a standard solution containing 0.1% ascorbic acid.

#### 2.3.7. Ethanol Levels

Juice ethanol concentrations were determined as described by Davis and Chace [25]. Generally, 10 mL aliquots of juice were incubated at 37 °C for 30 min in 25 mL Erlenmeyer flasks. In parallel, Erlenmeyer flasks containing 10 mL of 100 µL L^−1^ ethanol were used as internal standards for quantity evaluations. After the incubation, 2 mL gas samples were withdrawn from the Erlenmeyers’ headspaces into syringes and ethanol levels were determined with a Varian 3300 gas chromatograph. The results are means of three replications; each replicate included juice collected from three different fruits.

#### 2.3.8. Flavor

Flavor evaluations were conducted by three trained judges, who used a 1–9 hedonic scale, in which 1 = very bad and 9 = excellent.

#### 2.3.9. Acceptance Scores

Visual and final acceptance scores were assigned by three trained judges using a 5-grade scale, in which 1 = very bad, 2 = poor, 3 = fair, 4 = good and 5 = excellent.

### 2.4. Statistical Analysis

Feature-importance values were analyzed using mean Shapley additive explanation (SHAP) values [26] and the mean decrease accuracy (MDA) method [27]. The ClusViz tool was used for heatmap graphical representation [28]. Pearson correlation values were calculated using the open-source R software (available from: http://www.r-project.org; accessed on 10 April 2022).

### 2.5. Quality-Prediction Models

#### 2.5.1. Data-Set Preparation

The experiment produced 400-labeled data points; each data point represented one carton in the experiment. The input features were two pre-harvest variables (harvest date and yield) and three postharvest variables (storage time, temperature, humidity). The main model’s output (i.e., label) was the 5-grade scale final acceptance score assigned to each carton.

#### 2.5.2. Prediction Models

We tested four different linear and non-linear regression models for their ability to predict fruit-acceptance scores. These models are described below.

Multiple Linear Regression (MLR)—This basic model attempts to establish a linear relationship between the features and the label [29]. The MLR uses two or more independent features to predict the outcome of a dependent output label by fitting a hyperplane. The model finds the optimal parameters that minimize the mean squared error for the predicted quality scores.

Support Vector Regression (SVR)—SVR is a generalization of support vector machine (SVM) for regression tasks [30,31]. Unlike other models, SVR attempts to predict the label within a small range of allowed error. In other words, while MLR punishes every prediction error, SVR tolerates small errors as long as they fall within a predefined range. SVR models often employ kernels, which enable non-linearity in the input space. Non-linearity is achieved by transforming the data to a higher dimensional space, in which the relation between the inputs and label will be a linear one. In this work, a radial basis function (RBF) kernel was found to produce the best results, hence the results below relate to the use of SVR with an RBF kernel.

Random Forests (RF)—RF is a supervised ensemble method that is widely used for regression problems [32]. The RF model employs multiple regression trees (i.e., forests) to reduce the variance error [33]. For each tree, the model introduces different subsets of samples and features with replacements, also known as the bagging approach [34]. At prediction time (inference), each individual tree predicts a different value and the average of all of the predictions is used. The tree structure enables non-linearity and by averaging multiple predictions of different trees, RF often produces more accurate results than SVR or MLR.

Extreme Gradient Boosting (XGBoost)—This is a state-of-the-art ensemble method that has become popular in recent years for tabular data predictions [35]. Similar to RF, this model is based on regression trees. However, unlike RF, it uses a boosting approach instead of bagging. In boosting, the trees are built sequentially, with each tree trying to minimize the remaining error of all previous trees [36].

#### 2.5.3. Evaluation of the Models

A great deal of laboratory work was done to evaluate the 12,000 oranges and produce a data set of 400-labeled points. However, machine-learning models usually require many more data to learn and make predictions [37]. The common 80–20% train-set test-set split is risky when the data set is very small, since it is more likely to introduce bias. Hence, a K fold cross-validation method was used [38], with five folds and 20 repetitions producing 100 samples. If needed, pre-processing parameter scaling was applied to each iteration separately. Two metrics were used for the model evaluations: root mean squared error (RMSE) to compare the competing alternatives and the coefficient of determination (R^2^) to measure the amount of variance explained by each model.

#### 2.5.4. Duplication as a Way to Deal with Unbalanced Data Sets

In this study, the target label (i.e., acceptance score) is a 5-grade scale variable, with 5 denoting that the produce is very suitable for marketing and 1 denoting produce unlikely to be purchased. The advanced FEFO logistics strategy is expected to enable an efficient inventory management based on shelf-life predictions, and thus has great interest in predicting low-scoring fruit. However, low scores were relatively rare in the current data set, as only 14.75% of the samples in the total data set had scores of 3 (“fair”) or less. To cope with this challenge, training set samples with scores that were equal or lower than 3.25 were duplicated. Overall, six modes of duplication were compared: no duplication (i.e., 0) and 1 to 5 duplications.

## 3. Results

### 3.1. Effects of Pre-Harvest and Postharvest Features on the Quality of ‘Rustenburg’ Navel Oranges

To collect the high-throughput phenotyping data required to develop shelf-life prediction models, we examined the effects of various pre-harvest and postharvest features on the quality of ‘Rustenburg’ navel oranges. The examined pre-harvest features included harvest time (with six different harvest times ranging from 21 February 2021 to 6 April 2021) and orchard yield, which ranged between 18 and 48 Ton/Hectare (Table 1). The examined postharvest features included storage temperature (2, 5 or 8 °C), RH level during storage (70, 90 or 95%) and duration of storage, with the fruits being evaluated weekly over a period of 20 weeks.

The effects of the different features on the postharvest storage performance of ‘Rustenberg’ navel oranges, including weight loss, firmness, color, decay, peel damage, CI and internal dryness, are presented in Figure 1.

It can be seen that weight loss gradually increased during storage under all of the examined conditions, but reached its highest levels following storage under low-RH conditions. Firmness levels remained more or less stable over the course of the postharvest storage period, but gradually decreased under low-RH conditions.

It is worth noting that the fruits from all of the orchards had a yellowish pale color (hue angle between 59–61°) following storage at the low temperature of 2 °C, a normal orange color (hue angle between 62–64°) following storage at 5 °C and a deep orange/reddish color (hue angle between 66–69°) following storage at 8 °C (Figure 1 and Figure 2a). It is also worth noting that CI symptoms developed only upon storage at the low temperature of 2 °C, whereas internal dryness symptoms developed mainly during storage at the moderate and higher temperatures of 5 °C and 8 °C (Figure 1 and Figure 2b,c).

The effects of the various pre-harvest and postharvest features on the biochemical composition of the juice of ‘Rustenburg’ navel oranges, including TSS, acidity and vitamin C and ethanol levels, as well as flavor acceptance, are presented in Figure 3. The TSS levels remained relatively stable during storage, whereas acidity and vitamin C levels decreased and ethanol levels increased during storage. Fruit flavor acceptability remained relatively high (flavor acceptance score ≥ 7.0, on a scale of 1 to 9) for 14–18 weeks for the different orchards and treatments, but then tended to decrease, doing so somewhat more rapidly at the higher storage temperature of 8 °C.

Overall, fruit-acceptance scores remained relatively high (acceptance score ≥ 4, on a scale of 1 to 5) for 14–19 weeks, depending on the various pre-harvest and postharvest features (Figure 4). It is worth noting that the fruits stored at 5 °C maintained high acceptance scores for longer periods when compared to the fruits stored at a lower temperature of 2 °C or a higher temperature of 8 °C.

To evaluate the importance of the various pre-harvest and postharvest features on the quality of ‘Rustenburg’ navel oranges, we used various statistical analysis tools, including SHAP and RF MDA methods (Figure 5a,b). A feature importance analysis using mean SHAP values revealed that storage time was the most important feature affecting fruit quality, followed by storage temperature, harvest time, yield and humidity conditions (Figure 5a). Furthermore, RF MDA analysis revealed that harvest time had major effects on peel damage and vitamin C levels, and that orchard yield had major effects on decay, firmness and TSS. Regarding the postharvest features, storage time had major effects on ethanol accumulation, fruit flavor and acceptance scores; storage temperature had major effects on CI and peel color; humidity conditions during storage had a major effect on fruit weight loss (Figure 5b). It is worth noting that the different pre-harvest and postharvest features affected different postharvest quality parameters. For example, storage time mainly affected flavor, ethanol levels and acceptance scores, whereas storage temperature mainly affected color and CI (Figure 5b).

ANOVA revealed that the harvest date significantly affected acidity, vitamin C levels, decay, flavor, firmness, weight loss, peel damage, color and final acceptance scores, and orchard yields significantly affected acidity, vitamin C, flavor, weight loss and peel damage (*p* < 0.001; Table 2). Among the postharvest features, storage time significantly affected all of the examined parameters, with the exception of TSS. Storage temperature significantly affected all the parameters except for decay and TSS, and storage humidity significantly affected firmness and weight loss (Table 2).

Pearson correlations between the various quality parameters revealed positive correlations among flavor, visual and final acceptance scores and acidity and vitamin C levels, as well as negative correlations between these attributes and changes in ethanol levels, weight loss, decay, internal dryness, peel damage, color and CI (Figure 6; red and blue boxes, respectively). We also observed positive correlations between ethanol content, decay, weight loss and internal dryness, as well as negative correlations between weight loss and firmness (Figure 6; pink and black boxes, respectively).

Finally, Pearson correlations revealed positive correlations between final acceptance scores and acidity, vitamin C, flavor and visual acceptance scores, and negative correlations between final acceptance scores and changes in TSS, firmness, color, peel damage, decay, CI, internal dryness, weight loss and ethanol accumulation (Figure 7).

### 3.2. Quality-Prediction Models

Four regression models, including one linear model (MLR) and three non-linear models (SVR, RF and XGBoost), were evaluated for their abilities to predict acceptance scores based on two pre-harvest and three postharvest features (Table 3). Since the data set was not balanced, the models were evaluated twice. First, we evaluated the entire test sets (referred to as the “full set”) and then we evaluated a subset of the full set that consisted only of samples with acceptance scores equal or lower than 3.25 (i.e., low-quality samples, referred to here as the “low-score subset”). RMSE and R^2^ were calculated using a cross-validation method, which produced 100 samples, and the Wilcoxon non-parametric signed-rank test was applied to the results. As shown in Table 3, the XGBoost regression model outperformed the other models (*p* < 0.01). The observed RMSE values of the XGBoost model for the full set and the low-score subset were 0.195 and 0.380, respectively; and the R^2^ values were 0.914 and 0.305, respectively. As expected, the RMSE and R^2^ model prediction values were higher for the full set than for the low-score subset.

To further improve the XGBoost prediction model, especially for the low-score subset, we used a duplication approach in which low-quality samples in the training set were duplicated for each repetition and fold. Overall, six duplication modes were compared: no duplication (i.e., zero) and one to five duplications. The RMSE and R^2^ of the full set and low-score subset were measured again for each duplication mode (Table 4). The duplications had a relatively small diminishing effect on the RMSE and R^2^ of the full set (RMSE increased from 0.195 to 0.217; R^2^ decreased from 0.914 to 0.891), but substantially reduced the RMSE of the low-score subset from 0.380 to 0.329 and increased the R^2^ of the low-score subset from 0.305 to 0.479.

The FIFO logistic management method is based on the notion that storage time is the most crucial feature for predicting fruit quality. This was supported by the SHAP analysis, which revealed that storage time was the most important feature affecting the postharvest quality of oranges (Figure 5a). However, the results suggest that the other pre-harvest and postharvest features had additional effects on fruit quality. To evaluate the relative importance of the other examined pre-harvest and postharvest features in the prediction model, we conducted post hoc analyses to explore the contributions of various feature subsets on fruit-quality predictions. The data subgroups included: (1) storage time only, (2) storage time and pre-harvest features (harvest time and yield), (3) storage time and postharvest features (temperature and humidity) and (4) storage time and all pre-harvest and postharvest features (harvest time, yield, storage time, storage temperature and storage humidity).

XGBoost models with five duplications of the training sets were used to compare these subgroups. RMSE and R^2^ values for the full set and the low-score subset are presented in Table 5. For the full data set, the addition of either pre-harvest (Subgroup 2) or postharvest data (Subgroup 3) lowered the RMSE from 0.371 (Subgroup 1; i.e., storage time) to 0.304 and 0.340, respectively, and increased the R^2^ from 0.687 (Subgroup 1), based on storage time alone, to 0.790 (Subgroup 2) and 0.739 (Subgroup 3). The inclusion of all features (Subgroup 4) further reduced RMSE to 0.217 and increased R^2^ to 0.891. For the low-score subset, the neither did the addition of either pre-harvest (Subgroup 2) or postharvest data (Subgroup 3) improve the RMSE and R^2^ values. However, we observed an improved RMSE of 0.329 and R^2^ of 0.479 when all of the pre-harvest and postharvest features were included in the model (Subgroup 4), relative to storage time alone (Subgroup 1), with an RMSE of 0.473 and an R^2^ of −0.074.

## 4. Discussion

The main goal of the current study was to conduct a large-scale, high-throughput phenotyping analysis of the effects of various pre-harvest and postharvest features on the quality of ‘Rustenburg’ navel oranges, in order to develop quality-prediction models to enable the implementation of the more efficient FEFO logistic management system [14,15].

Performing the current postharvest storage evaluations of ‘Rustenburg’ navel oranges yielded two main outcomes: (1) they contribute to our current understanding of the importance of various pre-harvest and postharvest features on the quality of navel oranges, and (2) they allowed for the generation of quality-prediction models for oranges.

The SHAP analysis revealed that storage time is the most important feature affecting the postharvest quality of oranges, but also showed that other pre-harvest and postharvest features also have meaningful effects on fruit quality (Figure 5a). Furthermore, feature-importance analysis conducted using RF MDA revealed that different pre-harvest and postharvest features affect different aspects of fruit quality. For example, storage temperature mainly affects color and CI, whereas RH mainly affects weight loss (Figure 5b). More specifically, ANOVA revealed that the harvest time, storage time and storage temperature features significantly affected most examined fruit quality parameters, whereas tree yield significantly affected acidity, vitamin C, flavor, weight loss and peel damage, and humidity significantly affected fruit firmness and weight loss (Table 2). Thus, we conclude that all of the examined pre-harvest and postharvest features eventually influence the postharvest quality of navel oranges and that the most comprehensive quality-prediction models will include as many factors as possible.

Similar findings regarding the importance of both pre-harvest and postharvest features for shelf-life predictions were recently reported for nectarines. That work found that pre-harvest features, such as irrigation methods and fruit load, and postharvest features, such as storage temperature and storage RH, all have major effects on the postharvest quality of nectarines [39]. Other studies involving the development of shelf-life prediction models for strawberries, rocket leaves and mushrooms have reported that storage temperature is the most important feature affecting shelf life and quality [40,41,42]. However, another study pointed to the importance of RH for preserving the postharvest quality of strawberries [43].

The calculation of Pearson correlations between various orange fruit-quality parameters and final acceptance scores revealed that the fruit quality parameters that were positively correlated with high acceptance scores were acidity, vitamin C level and flavor, and that these parameters were also positively correlated with one another (Figure 6 and Figure 7). In contrast, changes in firmness, weight loss, decay, internal dryness, peel damage, CI, ethanol accumulation and fruit color were all negatively correlated with the fruit acceptance scores (Figure 6 and Figure 7).

The second main goal of the current study was to develop fruit quality prediction models. To that end, we examined four regression models and found that RF and XGBoost provided very effective quality predictions when the full experimental data set was used. The XGBoost regression model had a low RMSE value of just 0.195 for an acceptance score scale of one to five, as well as a high R^2^ value of 0.914 (Table 3). A more or less similar RMSE value of 0.184 and an R^2^ value of 0.911 were recently reported for the development of a shelf life prediction model of table grapes using an optimized radial basis function (RBF) and neural network [16]. However, unfortunately, the collected data were not fairly balanced, as low acceptance scores (equal or below three on a scale of one to five) accounted for only 14.75% of the total data set and, accordingly, the observed RMSE and R^2^ values for the low-scoring data set were only 0.380 and 0.305, respectively. Thus, the tested regression models provided much better quality predictions for fruits with higher acceptance scores than for fruits with lower acceptance scores.

To address this obstacle, we added a further duplication of the low-score subset and found that doing so helped to reduce the observed RMSE from 0.380 to 0.329 and increased the R^2^ from 0.305 to 0.479 (Table 4). Nonetheless, a further improvement of the quality predictions of low-acceptance-score fruits will require the performance of additional experiments and the generation of larger data sets of fruit with low acceptance scores.

Finally, post hoc analyses revealed that quality prediction models based on storage-time data alone were much less accurate than quality prediction models based on storage time together with the other examined pre-harvest and postharvest features (Table 5). Thus, the development of accurate quality-prediction models required for the implementation of the FEFO intelligent logistic management system necessitates the collection and use of as much pre-harvest and postharvest data as possible.

## 5. Conclusions

The main objective of the current research was to conduct a large-scale, high-throughput phenotyping analysis of the postharvest storage performance of ‘Rustenburg’ navel oranges, in order to develop shelf-life prediction models to enable the implementation of the FEFO logistic management method. The key findings were that the storage time appeared to be the most important feature affecting fruit quality, but also other features, including harvest time, storage temperature, yield and humidity, majorly influenced fruit postharvest quality. Based on the achieved data, we examined the efficacy of different regression models for their ability to predict fruit quality, and found that using XGBoost combined with a duplication approach provided the most effective approach, and allowed the prediction of fruit-acceptance scores among the full data set, with an RMSE of 0.217 and an R^2^ of 0.891. The development of accurate shelf-life prediction models should now allow the implementation of the FEFO logistic management system for more efficient inventory management and reduction of losses.

## Figures and Tables

**Figure 1 foods-11-01840-f001:**
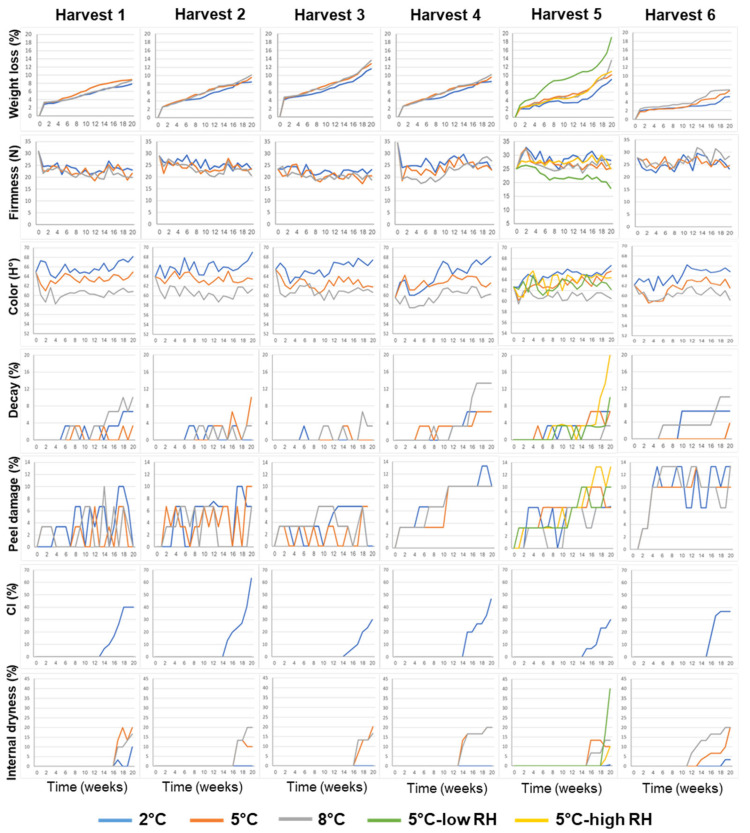
Effects of various pre-harvest and postharvest features on the postharvest storage quality of ‘Rustenburg’ navel oranges harvested from six different orchards. The harvest times and yields of the different orchards are presented in Table 1. The postharvest features included storage temperature (2, 5 or 8 °C) and storage RH level (70, 90 or 95%). Fruit quality was evaluated every week after one additional week of storage under shelf conditions (at 20 °C) for a period of 20 weeks.

**Figure 2 foods-11-01840-f002:**
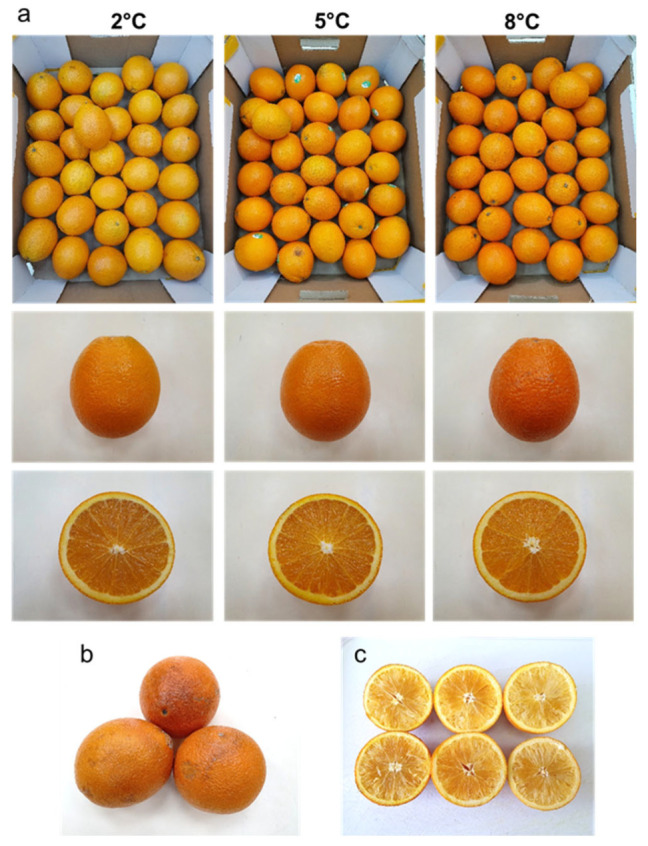
Photographs of ‘Rustenburg’ navel oranges. (**a**) Fruit stored at 2, 5 and 8 °C, (**b**) chilling injury, and (**c**) internal dryness symptoms.

**Figure 3 foods-11-01840-f003:**
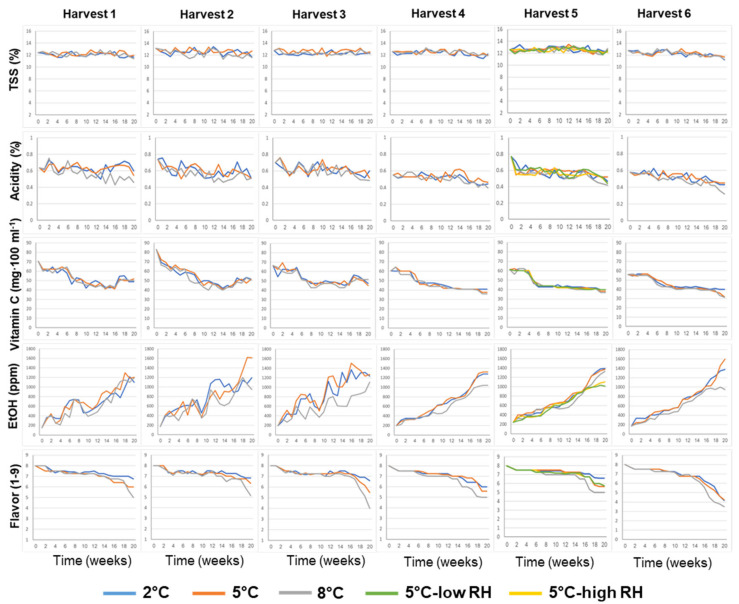
Effects of various pre-harvest and postharvest features on the biochemical composition and flavor of ‘Rustenburg’ navel oranges harvested from six different orchards. The harvest times and yields of the different orchards are presented in Table 1. The postharvest features included storage temperature (2, 5 or 8 °C) and storage RH (70, 90 or 95%). Fruit quality was evaluated every week after one additional week of storage under shelf conditions (20 °C) for a period of 20 weeks.

**Figure 4 foods-11-01840-f004:**
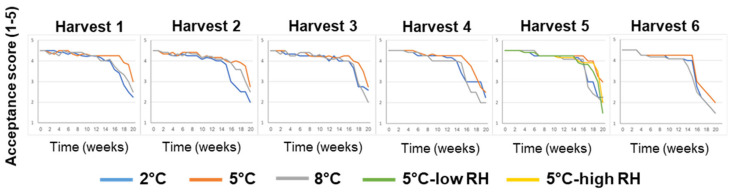
Effects of various pre-harvest and postharvest features on the acceptance scores of ‘Rustenburg’ navel oranges harvested from six different orchards. The harvest times and yields of the different orchards are presented in Table 1. The postharvest features included storage temperature (2, 5 or 8 °C) and storage RH (70, 90 or 95%). Fruit quality was evaluated every week after one additional week of storage under shelf conditions (at 20 °C) for a period of 20 weeks.

**Figure 5 foods-11-01840-f005:**
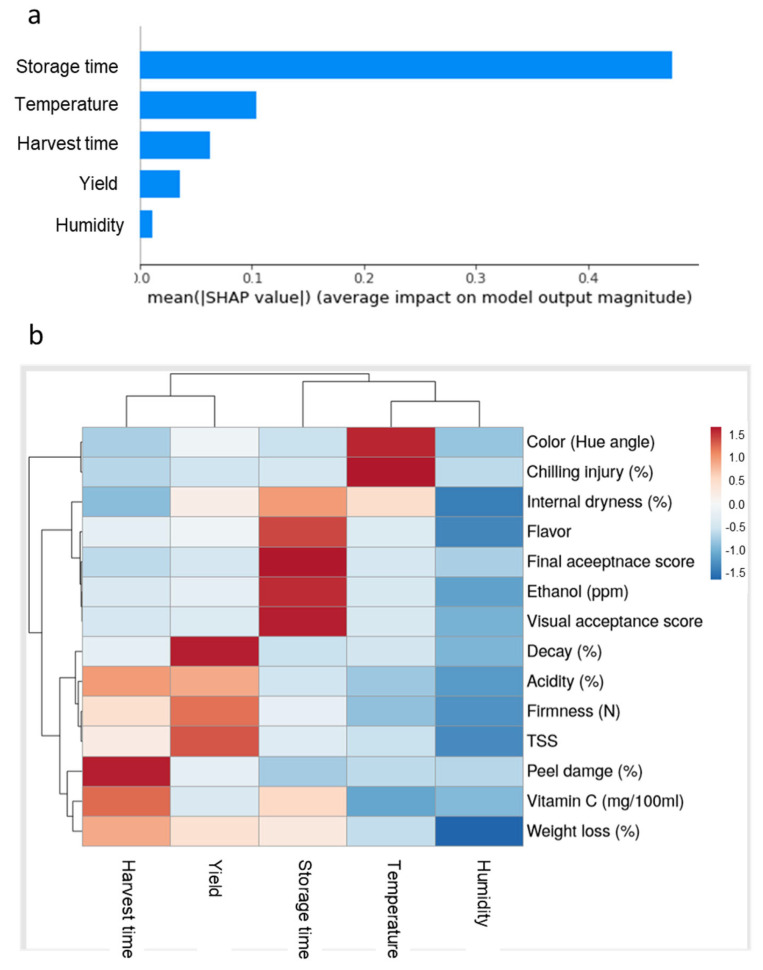
Importance of the various pre-harvest and postharvest features on the quality of ‘Rustenburg’ navel oranges. (**a**) Feature importance analysis using the SHAP method. (**b**) Feature importance and heat-map analyses using the RF MDA method.

**Figure 6 foods-11-01840-f006:**
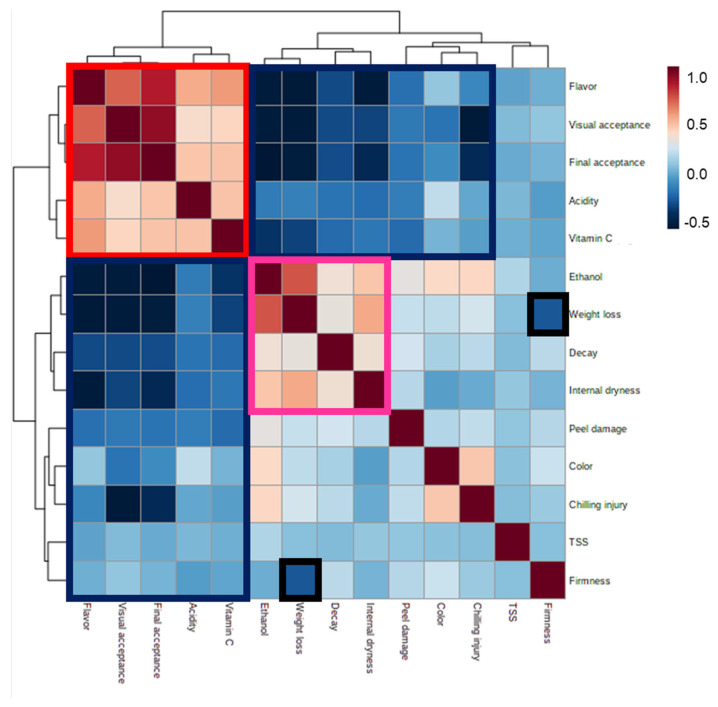
Pearson correlations and heat-map analysis of the various fruit-quality parameters. Red and pink boxes indicate parameters that are positively correlated with one another, and blue and black boxes indicate parameters that are negatively correlated with one another.

**Figure 7 foods-11-01840-f007:**
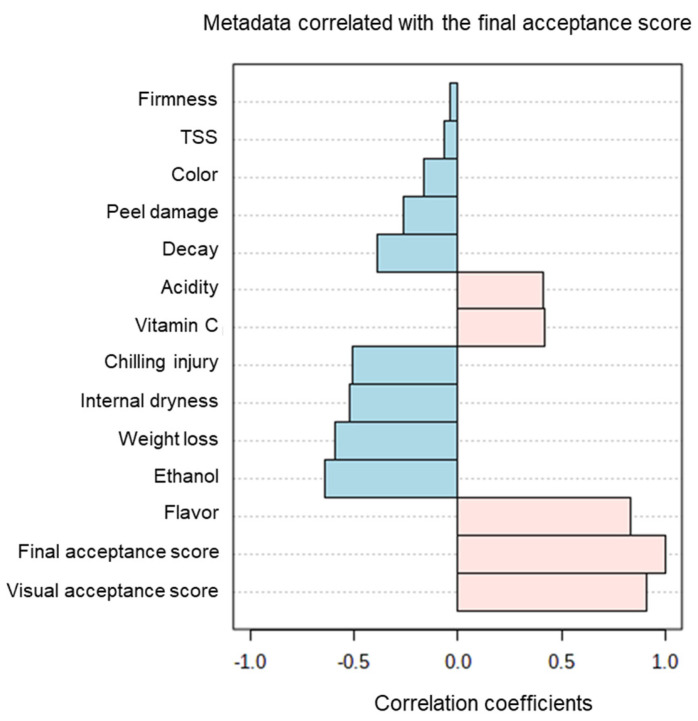
Pearson correlations between the various fruit-quality parameters and the final acceptance scores.

**Table 1 foods-11-01840-t001:** Harvest times and yields for the six different orchards used for the current experiment.

	Harvest Time (Weeks from Blooming)	Yield (Ton/Hectare)
Harvest 1 (21 February 2021)	48	48
Harvest 2 (24 February 2021)	48	44
Harvest 3 (28 February 2021)	49	18
Harvest 4 (17 March 2021)	51	35
Harvest 5 (25 March 2021)	52	18
Harvest 6 (6 April 2021)	54	33

**Table 2 foods-11-01840-t002:** Effects of different pre-harvest and postharvest features on the quality attributes of ‘Rustenburg’ navel oranges. The presented data are ANOVA *p* values. Gray shading indicates statistical significance (*p* ≤ 0.001).

	Harvest Time	Storage Time	Yield	Storage Temperature	Humidity
Acidity	6.55 × 10^42^	3.06 × 10^33^	2.01 × 10^10^	9.42 × 10^15^	-
Vitamin C	3.15 × 10^44^	4.20 × 10^210^	1.54 × 10^13^	7.17 × 10^17^	0.001
Internal dryness	-	6.98 × 10^106^	0.002	1.08 × 10^13^	-
Decay	4.86 × 10^6^	2.97 × 10^35^	0.001	0.034	0.005
Flavor	7.92 × 10^17^	4.37 × 10^268^	3.55 × 10^5^	3.08 × 10^16^	-
Final acceptance score	2.82 × 10^7^	0	0.003	3.03 × 10^7^	-
Firmness	7.56 × 10^69^	5.58 × 10^9^	-	5.11 × 10^16^	2.07 × 10^21^
Ethanol	0.004	0	0.002	1.76 × 10^17^	-
Visual acceptance score	-	1.47 × 10^279^	0.035	1.09 × 10^10^	-
Weight loss	2.33 × 10^27^	4.82 × 10^270^	1.58 × 10^16^	3.77 × 10^22^	6.08 × 10^12^
Peel damage	3.24 × 10^31^	3.95 × 10^16^	8.33 × 10^6^	3.28 × 10^6^	0.019
Hue angle	3.47 × 10^8^	4.15 × 10^12^	0.001	2.33 × 10^238^	0.006
TSS	-	-	-	-	-
Chilling injury	-	1.46 × 10^44^	0.030	5.12 × 10^45^	0.012

**Table 3 foods-11-01840-t003:** RMSE and R^2^ values for the MLR, SVR, RF and XGBoost regression models for the full data set and the low-score subset.

	Full Set		Low-Score Subset	
Algorithm	RMSE	R^2^	RMSE	R^2^
MLR	0.444	0.566	0.831	−2.314
SVR	0.312	0.784	0.662	−1.107
RF	0.200	0.910	0.404	0.217
XGBoost	0.195	0.914	0.380	0.305

**Table 4 foods-11-01840-t004:** RMSE and R^2^ values for the XGBoost regression model for the full data set and the low-score subset with 0 to 5 duplications.

	Full Set		Low-Score Subset	
Number of Duplications	RMSE	R^2^	RMSE	R^2^
0	0.195	0.914	0.380	0.305
1	0.201	0.907	0.358	0.385
2	0.206	0.903	0.344	0.433
3	0.210	0.898	0.337	0.456
4	0.214	0.894	0.333	0.469
5	0.217	0.891	0.329	0.479

**Table 5 foods-11-01840-t005:** The RMSE and R^2^ values for the XGBoost regression model with five duplications for the full data set and low-score subset, using four different feature subgroups.

	Full Set		Low-Score Subset	
Subgroup	RMSE	R^2^	RMSE	R^2^
1. Storage time	0.371	0.687	0.473	−0.074
2. Storage time + pre-harvest features (harvest time, yield)	0.304	0.790	0.464	−0.036
3. Storage time + postharvest features (temperature, humidity)	0.340	0.739	0.535	−0.376
4. Storage time + pre-harvest (harvest time, yield) + postharvest features (temperature, humidity)	0.217	0.891	0.329	0.479

## Data Availability

The data presented in this study are available on request from the corresponding author.

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
