# Peer review of "Large-Scale, High-Throughput Phenotyping of the Postharvest Storage Performance of ‘Rustenburg’ Navel Oranges and the Development of Shelf-Life Prediction Models"

_foods, 2022, doi:10.3390/foods11131840_

Round 1

Reviewer 1 Report

1. For the method section, it will be good to explain the rational for choosing the  difference harvest maturity indices used for harvesting at different farms

2. Having measured the colour it would be better if colour was described numerically instead of stating "all were pale yellow". 

3. Fig 1,3 and 4 should be made bigger for clarity. Not easy to see the results

Author Response

1. For the method section, it will be good to explain the rational for choosing the  difference harvest maturity indices used for harvesting at different farms – we now indicated that the rational for choosing different harvest times was to examine the postharvest performances of fruit with different maturity indices, i.e. different initial total soluble solids (TSS), acidity, peel color, etc. (p. 2-3, lines 96-99).

2. Having measured the colour it would be better if colour was described numerically instead of stating "all were pale yellow". – we now indicated the peel color in hue angle numerical values as suggested (p. 7, lines 267-269).

3. Fig 1,3 and 4 should be made bigger for clarity. Not easy to see the results – we now increased the sizes of Figs. 1,3 and 4 as suggested (p. 6 and 8).

Reviewer 2 Report

The manuscript entitled “Large-scale, high-throughput phenotyping of postharvest storage performance of 'Rustenburg' navel oranges and the development of shelf-life prediction models” authored by Abiola Owoyemi, Ron Porat, Amnon Lichter, Adi Doron-Faigenboim, Omri Jovani, Noam Koenigstein, and Yael Salzer presents hugely interesting aspects of post-harvest storage performances of oranges. The experiments have been meticulously carried out and results have been compiled systematically.

Here, there are certain points I suggest to the authors to improve the manuscript.

1.      The authors are requested to thoroughly check the typographical mistakes throughout the manuscript. For example, Figure 5; Line 207-210, etc.

2.      The authors are requested to elaborate the discussion part. In the present draft conclusive points have been added under discussion title. In discussion part the authors are requested to include some relevant references and reasons for the observations obtained during the experiments. There are many literatures already available on this subject. And, this will help the readers to gain more understanding on this research.

3.       The conclusion section is missing in the manuscript. There are large number of variables taken into consideration for this study. So, it becomes complicated to understand the outcome of this study. The authors are requested to include key points to summarize the results of the study.

4.      I recommend a minor revision of the manuscript.

Author Response

1   1. The authors are requested to thoroughly check the typographical mistakes throughout the manuscript. For example, Figure 5; Line 207-210, etc. – we thoroughly checked the entire manuscript for typographical errors.

2. The authors are requested to elaborate the discussion part. In the present draft conclusive points have been added under discussion title. In discussion part the authors are requested to include some relevant references and reasons for the observations obtained during the experiments. There are many literatures already available on this subject. And, this will help the readers to gain more understanding on this research – we now elaborated the discussion chapter (p. 14, lines 424-428 and 452-455).

3. The conclusion section is missing in the manuscript. There are large number of variables taken into consideration for this study. So, it becomes complicated to understand the outcome of this study. The authors are requested to include key points to summarize the results of the study – we now added a new conclusions chapter summarizing the key results of this study (p. 14-15, lines 473-486).